# Time series analyses with psychometric data

Tatjana Stadnitski [ORCID] *

Department of Research Methods, Institute of Psychology and Education, Ulm University, Ulm, Germany

* tatjana.stadnitski@uni-ulm.de

**Data Availability Statement:** The data are generated in Monte Carlo simulations. R codes for generating the data are provided in the Supporting Information files.

**Funding:** The author(s) received no specific funding for this work.

## Abstract

Understanding of interactional dynamics between several processes is one of the most important challenges in psychology and psychosomatic medicine. Researchers exploring behavior or other psychological phenomena mostly deal with ordinal or interval data. Missing values and consequential non-equidistant measurements represent a general problem of longitudinal studies from this field. The majority of process-oriented methodologies was originally designed for equidistant data measured on ratio scales. Therefore, the goal of this article is to clarify the conditions for satisfactory performance of longitudinal methods with data typical in psychological and psychosomatic research. This study examines the performance of the Johansen test, a procedure incorporating a set of sophisticated time series techniques, in reference to data quality utilizing a Monte Carlo method. The main results of the conducted simulation studies are: (1) Time series analyses require samples of at least 70 observations for an accurate estimation and inference. (2) Discrete data and failing equidistance of measurements due to irregular missing values appear unproblematic. (3) Relevant characteristics of stationary processes can be adequately captured using 5- or 7-point ordinal scales. (4) For trending processes, at least 10-point scales are necessary to ensure an acceptable quality of estimation and inference.

## Introduction

Process-oriented methodologies have become increasingly popular in different fields of empirical research. Multivariate time series techniques that treat variables as an interacting dynamic system, revealing their internal dynamics, represent relevant tools for understanding behavior or other psychological phenomena. In the last decade, there are growing numbers of empirical studies in psychology and psychosomatic medicine exploring intra-individual variability by means of sophisticated time series techniques like vector autoregressions, cointegration or vector error correction [1–5]. The problem is that these methods originate from physics, engineering, or econometrics and therefore are primarily designed for equidistant data measured on ratio scales. In psychology or psychosomatic, however, researchers mostly deal with ordinal or interval data. Furthermore, the measurements are often non-equidistant and contain missing values. Thus, the goal of this article is to clarify the conditions for satisfactory performance of time series analyses with psychometric data. We approach this issue by evaluating the performance of the *Johansen test* [6, 7] in reference to the data quality by using a Monte Carlo

**Competing interests:** The authors have declared that no competing interests exist.

method. We chose the Johansen test for our simulation studies because its algorithm comprises various sophisticated time series techniques: it can handle autocorrelations; it is applicable to both stable and unstable data; it allows distinguishing between different types of dynamic systems; it provides the maximum likelihood estimation of time series parameters. Moreover, the Johansen test demonstrated good performance in Monte Carlo evaluations with continuous data [e.g., 8].

## Theoretical background

This part of the article briefly introduces basic concepts of uni- and multivariate time series analyses and presents the Johansen test as a method for distinguishing between different types of dynamic systems. Stadnitski and Wild (2019) provide a detailed description of the methods within psychosomatic research and demonstrate their implementation with the *R* software [9]. For comprehensive examples of process-orientated modeling with empirical data from psychological research, consult Stadnitski (2014) [10].

### Univariate time series models

To explain basic concepts of the time series approach, we start with a *first order autoregressive model*

$$y_t = \beta y_{t-1} + u_t \tag{1}$$

which is a subtype of an autoregressive model of order *m* abbreviated as AR(*m*): $y_t = \beta_1 y_{t-1} + \beta_2 y_{t-2} + \ldots \beta_m y_{t-m} + u_t$. The autoregressive model incorporates the past or lagged values of the dependent variable as predictors (i.e., dependent and independent variables are both endogenous, that is, determined and interrelated inside the organism or system). External influences can enter the autoregressive system exclusively through the term $u_t$. The residuals $u_t$ are not autocorrelated; this time-independent process is termed *white noise*. To determine external influences in the autoregression, the model from Eq 1 must be represented alternatively in the so-called *moving-average form*

$$y_t = \sum_{i=0}^{\infty} \beta^i u_{t-i} = u_t + \beta u_{t-1} + \beta^2 u_{t-2} + \ldots \tag{2}$$

Since all independent variables in the autoregression are endogenous (lagged values of the dependent variable), Eq 2 points to the fact that autoregression actually models the impact of external random influences on the long-term development of the system. Therefore, the regression coefficient β in Eqs 1 and 2 can be interpreted as a memory parameter: β = 0 implies that the series $y_t$ is uncorrelated white noise with no memory; |β|<1 means that the effect of a random shock dissolves quickly without significant long-term impact on time series characteristics such as level or variance (short memory). Consequently, such processes are stable and their autocorrelations decline exponentially. When β = 1, the effect of a particular impulse does not dissipate over time and the series remembers the shock forever (long memory); as a result, observations remain strongly correlated even if they are far apart in time. The autoregressive process with β = 1 is called *integrated* of order 1 abbreviated I(1). Accordingly, short memory processes with |β|<1 are termed I(0). The I(1) process exhibits unstable trending behavior due to stochastic drifts, its first difference, $\Delta y_t = y_t - y_{t-1}$,

$$\Delta y_t = (1 - \beta)y_{t-1} + u_t = \rho y_{t-1} + u_t \tag{3}$$

is white noise.

Stability or instability as well as memory characteristics of time series can be inferred from their *autocorrelation functions* (ACF): non-zero autocorrelations at only a few lags are typical for stable short-memory processes, whereas significant autocorrelations on many lags indicate long memory or instability. Stationarity tests like the *Augmented Dickey-Fuller (ADF)* algorithm provide further possibilities to explore this issue. *Stationarity* means that the statistical characteristics of a process under study do not change over time (e.g., exhibit no trends or distinct fluctuations of mean or variance). Most time series are non-stationary due to a deterministic time trend or stochastic drift. The most general testing equation of the ADF test incorporates a constant term $c$ for modelling drift, a coefficient $\tau$ for modelling time trend (T) and allows autocorrelated residuals $a_t$:

$$\Delta y_t = c + \tau T + \rho y_{t-1} + a_t. \tag{4}$$

The null hypothesis of the ADF test is $\rho = 0$. Please note, if $c$ as well as $\tau$ equal zero and the residuals are white noise, we obtain Eq 3 where the dependent variable is differenced and the predictors are not, the parameter $\rho$ can therefore not be interpreted as the autocorrelation coefficient $\beta$ in Eq 1: $\rho = 0$ means $\beta = 1$, which implies that the series is integrated ($y_t = y_{t-1} + u_t$), $\rho < 0$ stands for $|\beta| < 1$ –that is, $y_t$ is stationary. For an applied description of stationarity tests, see Stadnytska (2010) [11].

## Multivariate time series models

The simplest vector autoregressive VAR($m$) model incorporates two processes. In contrast to the univariate autoregression, the current values of each variable are predicted not only from their own $m$ lagged values; $m$ extraneous lags (from the other variable) also serve as predictors. For instance, the bivariate VAR(1) model consists of two equations, each of them includes two predictors ($y_{1,t-1}$ and $y_{2,t-1}$): $y_{1,t} = \beta_{11}y_{1,t-1} + \beta_{12}y_{2,t-1} + u_{1,t}$ and $y_{2,t} = \beta_{21}y_{1,t-1} + \beta_{22}y_{2,t-1} + u_{2,t}$ or in matrix notation

$$\begin{pmatrix} y_{1,t} \\ y_{2,t} \end{pmatrix} = \begin{pmatrix} \beta_{11} & \beta_{12} \\ \beta_{21} & \beta_{22} \end{pmatrix} \begin{pmatrix} y_{1,t-1} \\ y_{2,t-1} \end{pmatrix} + \begin{pmatrix} u_{1,t} \\ u_{2,t} \end{pmatrix} \rightarrow y_t = B y_{t-1} + u_t. \tag{5}$$

There are three types of dynamic systems with quite different characteristics: stationary, integrated and cointegrated. *Stationary systems* consist of stable processes without trends [e.g., $y_{1,t}$ and $y_{2,t}$ are I(0)]. *Integrated systems* incorporate integrated processes without common stochastic components [e.g., $y_{1,t}$ and $y_{2,t}$ are I(1)]. Systems of integrated time series with common trends so that they move together to some extend are called *cointegrated*.

Suppose that two processes share the same I(1) element $x_t$

$$y_{1,t} = b_1 x_t + u_{1,t} \qquad y_{2,t} = b_2 x_t + u_{2,t},$$

where $u_{1,t}$ and $u_{2,t}$ are stationary or I(0), then the following linear combination

$$y_{1,t} - \frac{b_1}{b_2} y_{2,t} = b_1 x_t + u_{1,t} - \frac{b_1}{b_2} b_2 x_t - \frac{b_1}{b_2} u_{2,t} = u_{1,t} - \frac{b_1}{b_2} u_{2,t}$$

is the weighted sum of stationary variables and therefore also I(0). Although the series $y_{1,t}$ and $y_{2,t}$ are individually integrated (that is, they have stochastic trends), there exists a stationary linear combination suggesting that the two variables have a long-term equilibrium relationship between them. The bivariate process of the previous example is called cointegrated of order 1 or CI(1). The relation $y_{1,t} = b_1/b_2 y_{2,t} \rightarrow y_{1,t} = \beta_0 y_{2,t} \rightarrow y_{1,t} - \beta_0 y_{2,t} = 0$ characterizes the long-run equilibrium between two processes. As noted earlier, time series can be represented

alternatively in a differenced form (e.g., Eqs 3 and 4). Processes of cointegrated bivariate VAR (1) systems have the following differenced presentation, which is termed *Vector Error Correction Model (VECM)*

$$\Delta y_{1,t} = \alpha_1 (y_{1,t-1} - \beta_0 y_{2,t-1}) + \gamma_{11,1}\Delta y_{1,t-1} + \gamma_{12,1}\Delta y_{2,t-1} + u_{1,t}$$

$$\Delta y_{2,t} = \alpha_2 (y_{1,t-1} - \beta_0 y_{2,t-1}) + \gamma_{21,1}\Delta y_{1,t-1} + \gamma_{22,1}\Delta y_{2,t-1} + u_{2,t}$$

$$\Delta y_t = \alpha\beta' y_{t-1} - \Gamma_1 \Delta y_{t-1} + u_t = \Pi y_{t-1} - \Gamma_1 \Delta y_{t-1} + u_t \text{ with } \Pi = \alpha\beta' := \begin{bmatrix} a_1 & -\beta_0 a_1 \\ a_2 & -\beta_0 a_2 \end{bmatrix} \quad (6)$$

where the cointegrating vector $\beta = (1, \beta_0)'$ models the long-run equilibrium relation between processes: $y_{1,t-1} - \beta_0 y_{2,t-1}$ is the change in both variables that depends on the deviation from the equilibrium in period $t-1$. The absolute values of $\alpha_1$ and $\alpha_2$ reflect how quickly the variables restore the equilibrium. The $\gamma$ coefficients of lagged differenced predictors capture short-term autocorrelations of the cointegrated system. For an elaborated discussion of cointegration in the context of psychological research, consult Stroe-Kunold et al. (2012) [12].

## Johansen test

Note that every VAR(m) process has the representation

$$\Delta y_t = \Pi y_{t-1} - \Gamma_1 \Delta y_{t-1} - \ldots - \Gamma_{m-1}\Delta y_{t-m+1} + u_t,$$

whereas the relation $\Pi = \alpha\beta'$ of the VECM representation holds for cointegrated processes only. The rank of the matrix $\Pi$ can therefore disclose properties of the system under study. For instance, in a stationary system all $k$ series are stationary, thus linear combinations of them remain stationary. Hence, we can construct $k$ independent stationary linear combinations which implies that $\Pi$ has a full rank: $rk(\Pi) = k$. In the absence of common trends neither linear combination of integrated series becomes stationary, therefore $rk(\Pi) = 0$. The decomposition of $\Pi$ as the product of two $k$ x $r$ matrices, $\Pi = \alpha\beta'$, is only possible for cointegrated data. The number of independent cointegrated relations ($r$) must be smaller than $k$ and depends on the amount of common stochastic trends in the system ($m$): $rk(\Pi) = r = k - m$. For example, the bivariate VAR(1) system consists of two processes ($k = 2$), hence maximal one equilibrium relation $r = 1$ is possible: $rk(\Pi) = 2$ means the system is I(0), $rk(\Pi) = 1$ stands for CI(1), and $rk(\Pi) = 0$ implies that both series are I(1) without a common trend. The basic objective of the *Johansen test* is to distinguish between the different types of dynamic systems by estimating the rank of the matrix $\Pi$. In addition, the Johansen procedure provides the maximum likelihood estimation of the parameters $\alpha$ and $\beta$ from Eq 6 for cointegrated systems.

Fig 1 demonstrates the performance of the Johansen test for simulated bivariate I(0), I(1), and CI(1) systems. We generated data using the *R* statistical environment and conducted the Johansen test with the command **ca.jo** of the *R* package **urca** [13]. All *R* commands of the discussed examples are provided in the **Appendix** in S1 File. Both time series of the I (0) example are stationary: $y_{1,t} = -0.5 y_{1t-1} + u_{1,t}$, $y_{2,t} = -0.2 y_{2t-1} + u_{2,t}$ ($|\beta| < 1$). The lag = 1 autocorrelations of the ACF provide estimations of $\beta$s. The ADF test performed with the command **adf.test** of the *R* package **tseries** rejects the $H_0$: $\rho = 0$ ($\beta = 1$) in both cases ($p_{ADF1} = .01$, $p_{ADF1} = .02$). The Johansen test rejects the null hypotheses $rk(\Pi) = 0$ and $rk(\Pi) = 1$ because the test statistics exceed the 1% level significantly (97.67>23.52; 27.54>11.65). The estimated rank of the matrix $\Pi$ is therefore 2. In the I(1) example the integrated series do not share a common trend. The ADF maintains the $H_0$: $\rho = 0$ in both cases ($p_{ADF1} = .79$, $p_{ADF1} = .21$). As expected, the Johansen test does not reject the null hypothesis $rk(\Pi) = 0$: the test statistic (13.12) is smaller as the

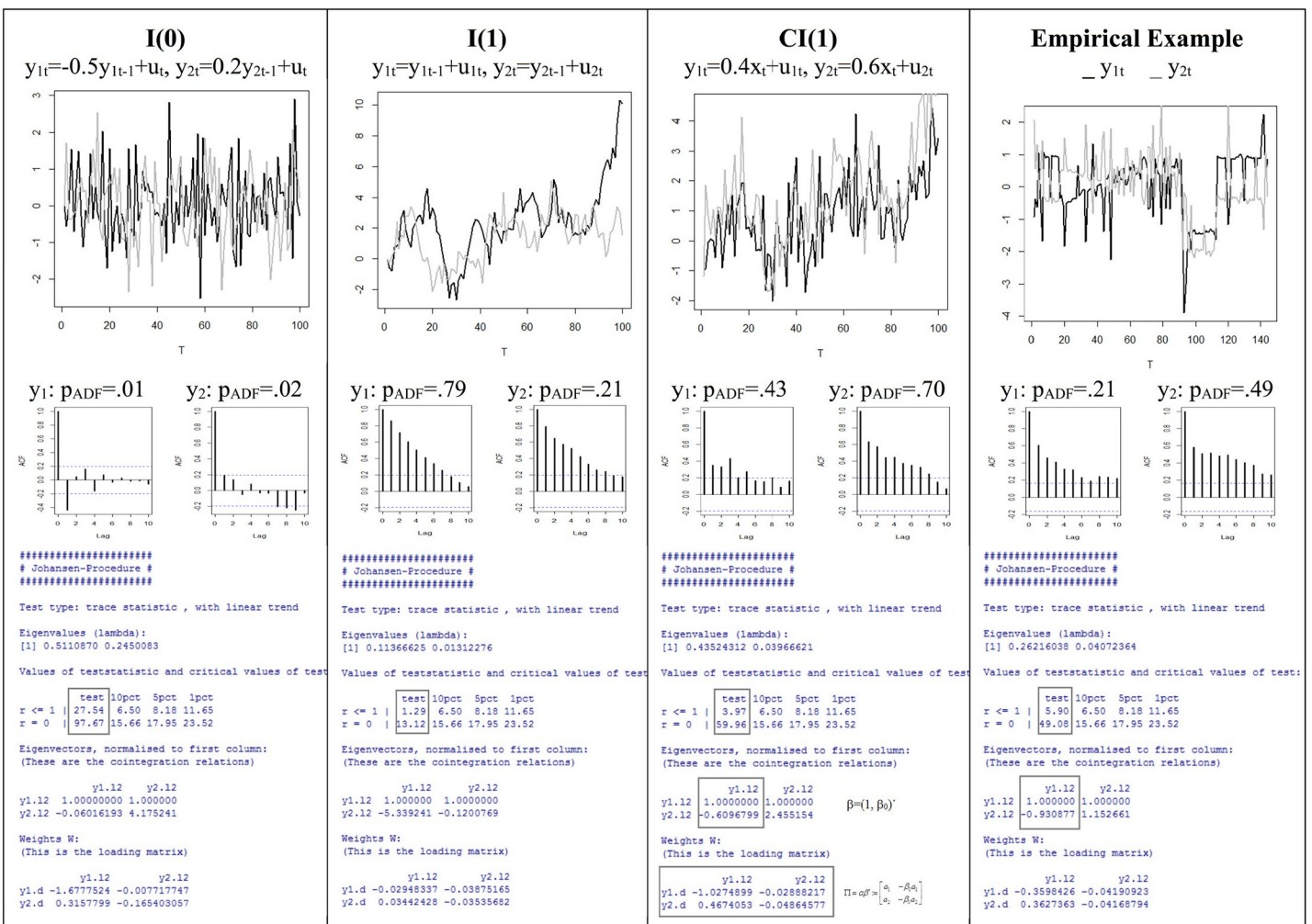

**Fig 1. Simulated bivariate I (0), I (1), CI (1) models and a bivariate empirical system with their autocorrelation functions and R outputs of the Johansen test.**

1% critical value (23.52). The CI (1) series $y_{1,t} = 0.4x_t + u_{1,t}$ and $y_{2,t} = 0.6x_t + u_{2,t}$ share the same I (1) element $x_t$, the cointegrating vector is $\beta = (1, -0.4/0.6 = -0.67)'$. The ADF identifies both cases as integrated ($p_{ADF1} = .43$, $p_{ADF1} = .70$). The estimated rank of the matrix $\Pi$ in the Johansen test is 1, since the $H_0$: rk $(\Pi) = 0$ is rejected (59.96>23.52) and $H_0$: rk $(\Pi) = 1$ is maintained (3.97<11.65). The estimated cointegrating vector is $(1, -0.61)'$. Thus, the following linear combination $y_{1,t} - 0.61y_{2,t}$ of two integrated series is stationary.

The data of the empirical example in Fig 1 originated from the study by Kupfer, Brosig, and Brähler (2005), in which the authors analyze the marital interaction of a married couple under clinical conditions over a period of 144 days [14]. The time series represent the mood of the couple measured on Likert scales, the measurements are standardized. According to the Johansen test, the series are cointegrated since the estimated rank of the matrix $\Pi$ is 1, the estimation of $\beta_0$ is -0.93. In contrast to the simulated data, however, the correctness of the decision remains uncertain. The following Monte Carlo simulations aim, among other things, in providing information about the probability for obtaining a correct test decision in such cases.

## Empirical versus simulated data

As stated above, the goal of this study is to clarify the conditions for satisfactory performance of the Johansen test with non-continuous data typical in psychological research. This goal can be achieved only with simulated data. To understand this statement, one should first know the differences amongst the following statistical concepts: "parameter", "estimator", and "estimate". A *parameter* is a quantity that defines a particular system, such as the mean of the normal distribution. Strictly speaking, to obtain a population parameter one must measure an entire population, which is mostly infeasible, so instead one generally uses *estimators* (rules or formulae) to infer population parameters from observed samples. For any parameter, there are usually multiple estimators with diverse statistical properties. As an example, suppose we have $n$ observations of some phenomenon X; then we can estimate the population mean ($\mu$) using two well-known estimators, the sample mean, $\hat{\mu}_1 = \frac{1}{n} \sum\limits_{i=1}^{n} X_i$, and the sample median, $\hat{\mu}_2 = X_{0.5}$. In contrast to parameters, estimators are not numbers but functions characterized by their distributions, expectancy values, and variances. An *estimate* is a particular numerical value obtained by applying an estimator. Good estimators are unbiased, i.e., their means equal the true parameter value, and have small variability, i.e., their estimates do not differ strongly. Considering that just one estimate per method is available in a typical research situation, an estimator with a narrow range is usually better than one with a broad range. For instance, both estimators of $\mu$ in large samples are normally distributed and unbiased; but $\hat{\mu}_1$ is a better estimator of $\mu$ than $\hat{\mu}_2$ because its variance is considerably smaller.

The Johansen test is the estimator of the present study. Among other things, it estimates the cointegrating vector $\beta = (1, \beta_0)'$. Fig 1 shows that the estimate of $\beta_0$ of the simulated CI(1) system is -0.61. Since the actual parameter is known in this case ($\beta_0 = -0.4/0.6 = -0.67$), the estimated value can be compared with it to obtain the estimation error: $-0.67-(-0.61) = -0.06$. Therefore, working with simulated data allows to quantify the quality of the estimation. In contrast, the parameter value of the empirical system in Fig 1 is unknown, hence the difference between the estimate of $\beta_0$ (-0.93) and the true parameter is incalculable, which makes quantification of the estimation error impossible. The examples of Fig 1 point to the fact that an evaluation of the Johansen test performance is only possible using simulated data.

Generally, an empirical way to determine the quality of an estimator is by means of *Monte Carlo simulations*. For instance, computational algorithms can generate a population with a known parameter value, and repeated samples of the same size can be drawn from this population, e.g., 1000 CI(1) systems with $\beta_0 = -1$ and T = 100, then an estimator (e.g., the Johansen test) can be applied to the data, yielding 1000 estimates of the parameter $\beta_0$. For a good estimator, the variability of the estimates must be low with the mean or median near the true parameter value.

## Method

The following Monte Carlo experiments evaluate the performance of the Johansen test for bivariate time series systems measured on different scales with or without missing values. As quality indicators serve the percentage of false model identifications and the estimating precision of the cointegrating parameter $\beta_0$ from Eq 6. Mean, median, interquartile range (IQR), and the percentage of estimates outside the 1.5 interquartile range (% OUT) obtained from 1000 replications are used as accuracy indicators of parameter estimations. All computations are performed with the *R* software.

For clarity purposes, we first evaluated the performance of the Johansen test for continuous data in dependence of system type ((I(0), I(1), CI(1)), significance level (10%, 5%, 1%), time

series length (T = 10 to T = 500) and parametrizations (e.g., $\beta_0 = -b_1/b_2$ for cointegrated systems; in stationary and integrated systems $\beta_0 = 0$, $b_1 \neq b_2$, means that a bivariate system consists of time series with unequal variances). The first experiment with N = 1000 replications delivered the following outcome: The Johansen test achieved an acceptable discriminating performance with less than 5% of misclassification at the 1% level of significance in samples of at least 70 observations. The effect of the system type on the classification accuracy was stronger than the influence of parameterizations, best results were obtained for stationary systems. Parameter estimates were unbiased (mean ≈ median), the accuracy of estimation distinctly improved with growing sample size. Based on these results for continuous data, the following presentations are confined to T = 100, $\alpha = 1\%$, and the parameterization $\beta_0 = -1$.

Incomplete data sets represent a widespread problem of psychological research; missing values are particularly prevalent in longitudinal studies [15]. In time series, missing observations usually imply not only shortened samples but also distorted equidistance. To investigate the impact of failing equidistance due to missing values on the performance of the Johansen test, we made the data discrete (only whole numbers were used as time series values) and manipulated the percentages of missing values (from 10% to 30%) as well as the nature of failing observations (regular vs. random).

To investigate the effect of scaling on the performance of the Johansen test, we compared test decisions and quality of estimations in dependence of the levels of measurement. We created various interval and ordinal scales as described in detail by Baker, Hardyck and Petrinovich (1966) [16]. For both scales, we varied the number of points: 3 through 10. In the ordinal case, we also manipulated the interval size between levels: the interval size varied randomly, increased from the median, decreased from the median, or increased monotonically.

Additionally, we examined the impact of score limitations at the top or the bottom of a scale on the performance of the Johansen test. In longitudinal analyses, ceiling effects can cause incorrect model selections and biased parameter estimations [17]. To investigate the impact of ceiling or floor effects on the performance of the Johansen test we created different data by merging a portion of the marginal values in a 10-point scale. The merged proportions varied from 30% through 50% with the scales bounded above, below or on both sides. For instance, in a scale bounded above the values 7, 8, 9, and 10 were summarized to 7 limiting the range of the scale.

R codes for generating the data are provided in the files **Dataset I (0)**, **Dataset I (1)**, and **Dataset CI** in S2–S4 Files. For more examples and elaborated explanations how to simulate interval or ordinal data with missing values or with floor and ceiling effects, consult Gruber (2011) [18].

In sum, the described Monte Carlo studies generated the following types of psychometric data: continuous data, interval data with different points, ordinal data with different points, ordinal data with various distances between levels, interval and ordinal data with varying percentages of missing values and with equidistant and non-equidistant omissions, interval and ordinal data with scales bonded above, below or on both sides. There are two main goals of time series analysis: identifying the nature of the phenomenon represented by the sequence of observations (**model identification**) and predicting future values of the time series variable which requires an accurate **parameter estimation**. The present study examines the goal attainment in dependence of the data type.

## Results

Fig 2 shows that the Johansen test achieved an acceptable discriminating performance with less than 5% of misclassification in samples with at least 70 observations. The estimates of $\beta_0$

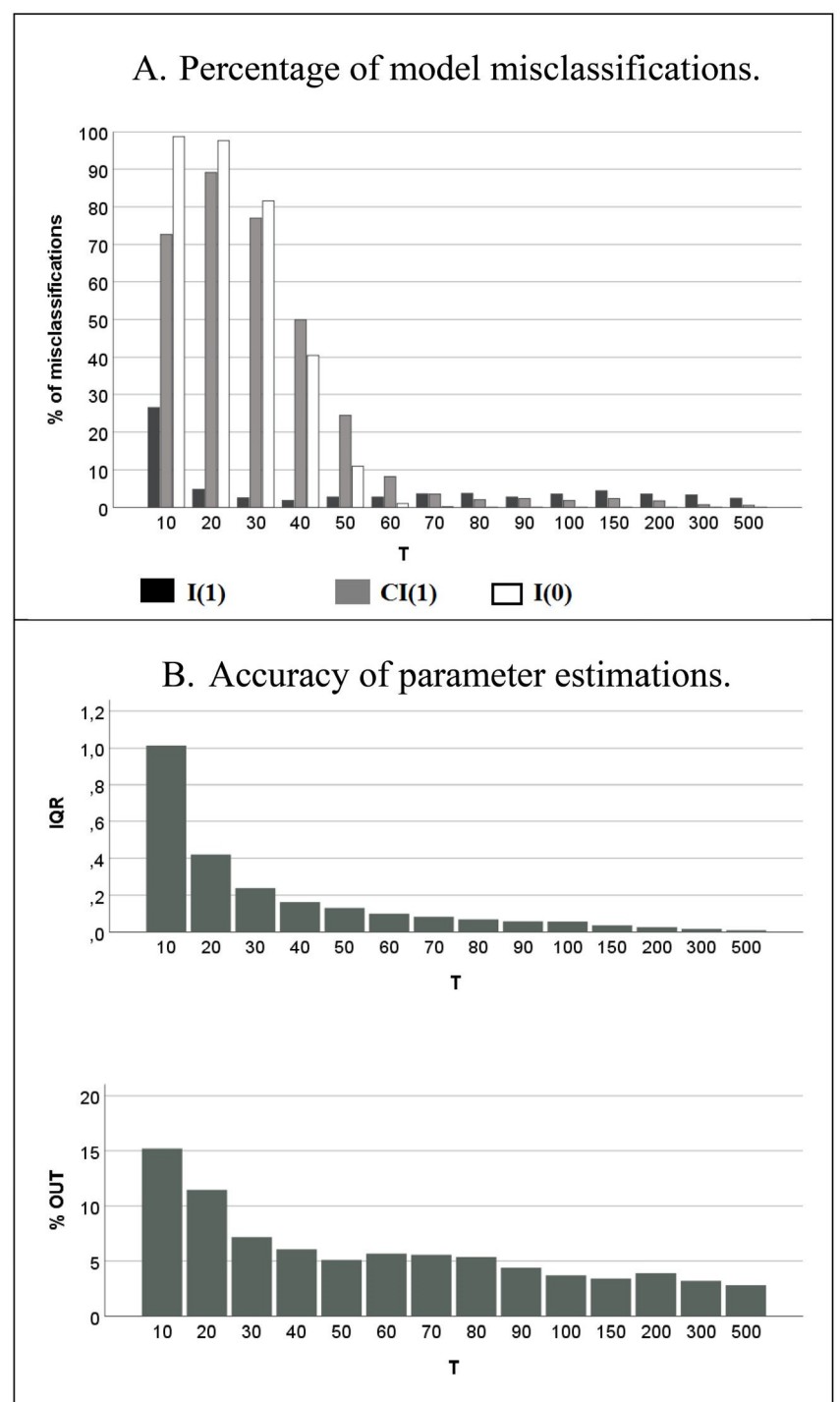

**Fig 2. Performance of the Johansen test in dependence of sample size.**

proved to be unbiased, therefore Fig 2 visualizes their variability in dependence of sample size and demonstrates a distinct improvement in the accuracy of the parameter estimation with increasing $T$. The results suggest $T = 100$ as a parsimonious optimal time series length.

Results presented in Table 1 show that in samples with 100 intended observations the Johansen test can cope with up to 30% of missing values: less than 5% of incorrect model

**Table 1. Performance of the Johansen test in time series with missing values.**

|  | % of misclassifications | | | Accuracy of estimation of $\beta_0 = -1$ | | |
| --- | --- | --- | --- | --- | --- | --- |
|  | I(1) | CI(1) | I(0) | MEDIAN | IQR | % OUT |
| complete data | 3.6 | 1.9 | 0 | −1.002 | 0.058 | 3.7 |
| every 10th value is missing | 3.2 | 1.6 | 0 | −1.003 | 0.060 | 3.6 |
| every 7th value is missing | 3.3 | 1.7 | 0 | −1.033 | 0.062 | 5.2 |
| every 5th value is missing | 2.6 | 1.6 | 0 | −1.035 | 0.058 | 4.0 |
| every 3rd value is missing | 2.5 | 4.2 | 0.4 | −1.147 | 0.071 | 4.8 |
| 10% random missing values | 3.2 | 1.8 | 0 | −1.005 | 0.056 | 4.3 |
| 14% random missing values | 3.2 | 1.9 | 0 | −1.004 | 0.061 | 3.8 |
| 20% random missing values | 1.9 | 1.5 | 0 | −1.002 | 0.063 | 4.6 |
| 30% random missing values | 1.9 | 4.3 | 0.2 | −1.002 | 0.072 | 4.2 |

identifications were observed; the estimations of $\beta_0$ were unbiased and precise. The test demonstrated similar discriminating and estimating quality in the regular and random cases. Therefore, the Johansen test does not necessarily require equidistant measurements for correct inference.

Table 2 summarizes the most important results concerning the levels of measurement. A 10-point scale seems to be necessary for a satisfactory discriminating accuracy with less than 10% of model misclassifications. For the ordinal scale, the number of points appeared to be more important than the cause for non-equidistant intervals, thus Table 2 provides summary statistics obtained from the data with different sizes between levels. The most common error was misidentification of integrated systems as cointegrated. Scales with more than 5 points were necessary for satisfactory performance of the Johansen test when dealing with processes containing stochastic trends. A 7-point ordinal scale was sufficient for accurate parameter estimation in the stationary case. Recall that the estimation of $\beta_0$ in cointegrated systems is based on a stationary linear combination.

Table 3 presents simulation results for the data with floor and ceiling effects. It demonstrates that even strong aggregation (50%) was not problematic for stationary processes. For merged proportions of no more than 30%, the test performance was even comparable to the performance based on untransformed continuous data. In systems with trending processes, however, the scale limitations were definitely disadvantageous for the performance of the Johansen test because ceiling and floor effects distinctly enhanced the number of incorrect model selections. Limited scales obviously failed to capture fluctuations of trending series adequately.

**Table 2. Performance of the Johansen test in time series measured on different scales.**

|  | % of misclassifications | | | Accuracy of estimation of $\beta_0 = -1$ | | |
| --- | --- | --- | --- | --- | --- | --- |
|  | I(1) | CI(1) | I(0) | MEDIAN | IQR | % OUT |
| continuous data | 3.6 | 1.9 | 0 | −1.002 | 0.058 | 3.7 |
| 10-point interval scale | 5.6 | 2.3 | 0 | −1.003 | 0.060 | 4.0 |
| 7-point interval scale | 8.9 | 2.2 | 0 | −1.001 | 0.067 | 3.7 |
| 5-point interval scale | 14.0 | 2.5 | 0 | −0.999 | 0.068 | 4.0 |
| 3-point interval scale | 29.1 | 6.8 | 0 | −1.000 | 0.082 | 4.5 |
| 7-point ordinal scale | 16.6 | 4.4 | 0 | −1.002 | 0.097 | 4.0 |
| 10-point ordinal scale | 8.2 | 3.3 | 0 | −1.003 | 0.076 | 3.5 |

**Table 3. Performance of the Johansen test in time series with floor and ceiling effects.**

| | % of misclassifications | | | Accuracy of estimation of $\beta_0 = -1$ | | |
|---|---|---|---|---|---|---|
| | **I(1)** | **CI(1)** | **I(0)** | **MEDIAN** | **IQR** | **% OUT** |
| untransformed data | 3.6 | 1.9 | 0 | −1.002 | 0.058 | 3.7 |
| scale is bounded above (30%) | 15.6 | 4.0 | 0 | −1.001 | 0.063 | 4.2 |
| scale is bounded below (30%) | 20.1 | 2.4 | 0 | −1.001 | 0.064 | 4.3 |
| scale is bounded above (50%) | 29.2 | 11.3 | 0 | −1.002 | 0.114 | 5.3 |
| scale is bounded on both sides (30%) | 16.4 | 1.8 | 0 | −1.000 | 0.046 | 5.1 |

## Summary and conclusions

Psychometric data are often "imperfect": measurements usually originate from ordinal scales like Likert questionnaires; moreover, missing observations are quite common. Consequently, psychological data from longitudinal designs are normally discrete values of a limited range with failing equidistance between them. The goal of the presented study was to find out under which conditions different sophisticated time series techniques implemented in the Johansen test work properly with data from empirical psychology.

The main results of the conducted Monte Carlo simulations are: (1) Time series analyses require samples of at least 70 observations for an accurate estimation and inference. (2) Discrete data and failing equidistance of measurements due to irregular missing values appear unproblematic. Thus, interruptions on weekends by investigating daily phenomena are non-hazardous and supplementary data collection in order to obtain an appropriate sample size is reasonable in longitudinal studies. (3) Relevant characteristics of stationary processes can be adequately captured using 5- or 7-point ordinal scales. (4) For trending processes, at least 10-point scales are necessary to ensure an acceptable quality of estimation and inference. Moreover, it is essential to consider a possible growth of the processes during scale construction, since ceiling or floor effects are especially consequential for series containing trends.

The results of the present study, among other things, allow for a better assessment of empirical findings from longitudinal empirical research. For instance, in the martial interaction study reported above, the Johansen test indicated that the mood time series of the married couple obtained on 144 successive days were cointegrated with the estimation of $\beta_0$ = -0.93. The mood measures originated from a questionnaire with 58 items like "*Right now I feel good*". The momentary intensity of emotions was rated with answers 1 = *definitely not*, 2 = *not*, 3 = *not really*, 4 = *a little*, 5 = *very much*, 6 = *extremely*. In summary, the data show the following characteristics: 6-point ordinal scale, T = 144, 0% missing values, standardized time series (i.e., with equal means and variances). In the Monte Carlo experiments, 6-point ordinal data with T≈150 and 7-point ordinal data with T = 100 provided similar results. Thus, from Table 2 follows that, under these conditions, the estimated probability of a correct identification for CI(1) systems is about 96%. On the other side, up to 16.6% I(1) systems can be misclassified as CI(1). The present study demonstrated that distinct ceiling or floor effects are expectable for 6-point ordinal data. Such scale limitations impede identification of cointegrated time series, i.e., forward misclassifications of CI(1) systems. Moreover, the estimate of $\beta_0$ = -0.93 from the empirical example indicates the following 1.5 interquartile range for $\beta_0$: [-0.93–1.5\*0.097; -0.93 +1.5\*0.097] = [-1.076; -0.785]. The findings from the present study suggest that, under the conditions described, CI(1) systems with $\beta_0$ = -1 are associated with the following 1.5 interquartile range for the estimations of $\beta_0$: [-1.002–1.5\*0.097; -1.002+1.5\*0.097] = [-1.148; -0.857]. The estimate from the empirical example (-0.93) lies inside the interval and is therefore probable for CI(1) data with $\beta_0$ = -1 (see % OUT in Table 2). Consequently, the analysis suggests that the evidence for cointegration is rather strong in this case.

The following examples demonstrate the implementation of the findings from the present Monte Carlo experiments in applied research. For this purpose, two clinical studies are used. The first study aimed at analyzing temporal relationships between awakening cortisol and psychosocial variables in inpatients with anorexia nervosa [4]. The items assessing psychosocial variables captured anticipations, depressive feelings, nervousness, anxiety, or stress (e.g., "*Today, I am starting the day with positive anticipations*", "*At the moment, I feel nervous*"). The goal of the second clinical study was to investigate the interaction between emotional intolerance and core symptoms of anorexia nervosa over the course of inpatient treatment [3]. Items as "*Today, I could not tolerate unpleasant emotions*" assessed emotional intolerance. The essential symptoms of anorexia nervosa were rated with items monitoring restraint over eating, weight concern, fear of losing control over eating, and preoccupation with food, e.g., "*Today, I had a definite fear of losing control over eating.*" Multivariate time series analyses require similar scaling for all variables within a system. Therefore, in the first study, psychosocial variables need a wide numeric scale to associate them appropriately with continuous cortisol measurements. Both studies deal with potentially trending phenomena since changes in mood or clinical symptoms during the treatment are probable. The present Monte Carlo simulations demonstrated that limited scales failed to capture fluctuations of trending series adequately. Moreover, they showed that at least 10-point scales are necessary to ensure an acceptable quality of estimation and inference in trending processes. Thus, to mimic a metric scale and to consider a possible growth of the processes during the scale construction, measurements of psychosocial variables, emotional intolerance, and symptoms of anorexia nervosa were obtained as follows: Patients rated each item on a visual analogue scale with bipolar labels. The marked points were converted by a computer program to a numeric scale, from 0 to 100, visible to the patient while completing the questionnaire. In both studies, data were collected daily. The present Monte Carlo experiments suggested that time series analyses require samples of at least 70 observations for an accurate estimation and inference. Furthermore, from the simulations follow that supplementary data collection to guarantee an appropriate sample size is reasonable in the presence of missing values. Therefore, clinical samples of both studies should primarily include patients with intended inpatient stay of at least three months.

## Supporting information

**S1 File. Appendix.**
(R)

**S2 File. Dataset CI.**
(R)

**S3 File. Dataset I (0).**
(R)

**S4 File. Dataset I (1).**
(R)

**S5 File. Empirical example.**
(CSV)

## Acknowledgments

The author would like to thank Antje Heinle (Gruber) for her support in developing R codes for Monte Carlo simulations.

## Author Contributions

**Conceptualization:** Tatjana Stadnitski.

**Writing – original draft:** Tatjana Stadnitski.

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
