## [Decision Letter · Decision Letter 0]

9 Jan 2020

PONE-D-19-32949

Time series analyses with psychometric data

PLOS ONE

Dear Dr. Stadnitski,

Thank you for submitting your manuscript to PLOS ONE. After careful consideration, we feel that it has merit but does not fully meet PLOS ONE’s publication criteria as it currently stands. Therefore, we invite you to submit a revised version of the manuscript that addresses the points raised during the review process.

We would appreciate receiving your revised manuscript by February 9, 2020. To enhance the reproducibility of your results, we recommend that if applicable you deposit your laboratory protocols in protocols.io, where a protocol can be assigned its own identifier (DOI) such that it can be cited independently in the future. For instructions see: http://journals.plos.org/plosone/s/submission-guidelines#loc-laboratory-protocols

We look forward to receiving your revised manuscript.

Kind regards,

Stephan Doering, M.D.

Academic Editor

PLOS ONE

Reviewers' comments:

Reviewer's Responses to Questions

**Comments to the Author**

1. Is the manuscript technically sound, and do the data support the conclusions?

Reviewer #1: Partly

Reviewer #2: Partly

2. Has the statistical analysis been performed appropriately and rigorously? 

Reviewer #1: I Don't Know

Reviewer #2: Yes

3. Have the authors made all data underlying the findings in their manuscript fully available?

Reviewer #1: Yes

Reviewer #2: Yes

4. Is the manuscript presented in an intelligible fashion and written in standard English?

Reviewer #1: No

Reviewer #2: Yes

5. Review Comments to the Author

Reviewer #1: The authors present a study in which they evaluate the performance of Johansen test as a function of the data quality (using a Monte Carlo approach). They apply different types of transformations on the data, e.g., missing values, changing the scale and floor/ceiling effects) and evaluate the effect on these. There is no technical novelty in the proposed approach, this is a sole experimental study. My main concern is the lack of more experiments to support their claims: the authors use only one dataset for the conducted experiments and expect that the conclusions will generalize on any type of psycometric data. Experiments with at least two additional (and diverse) datasets are expected to support the claims of the authors. Furthermore, there are several typos in the manuscript (e.g., "maxim likelihood estimation", l.49, "autregrssive", l.60). Finally, the actual contribution of the study is unclear: the authors should better highlight the contribution of the paper and demonstrate that this study will be indeed important for the scientist of the relevant field.

Reviewer #2: In this paper, authors present Time series analyses with psychometric data. This reviewer’s comments are as follows:

[1] Equation (2) seems incorrect or there is typo of term ut-1 as it should be ut-i

[2] It would be better if author can provide two-three real-life applications of the proposed study by considering details of actual problems and solutions of same in Results section.

6. PLOS authors have the option to publish the peer review history of their article (what does this mean?). If published, this will include your full peer review and any attached files.

Reviewer #1: No

Reviewer #2: Yes: Pushpendra Singh, NIT Hamirpur, HP, India

---

## [Author Response · Author response to Decision Letter 0]

1 Feb 2020

Reviewer #1 (1): Explanations that dozens of different data types were generated in the Monte Carlos simulations of the present study are provided (241-249).

Reviewer #1 (2): Typos are corrected.

Reviewer #1 (3): Contribution of the study is explained (309-360).

Reviewer #2 (1): Equation 2 is improved.

Reviewer #2 (2): Real-life applications are provided (309-360).

Editor (1): PLOS ONE's style requirements are checked.

Editor (2): A study's minimal data set is provided as supplementary files S2_File.R, S3_File.R, S4_File.R.

Editor (3): Figures are uploaded to the PACE.

---

## [Decision Letter · Decision Letter 1]

6 Mar 2020

PONE-D-19-32949R1

Time series analyses with psychometric data

PLOS ONE

Dear Dr. Stadnitski,

Thank you for submitting your manuscript to PLOS ONE. After careful consideration, we feel that it has merit but does not fully meet PLOS ONE’s publication criteria as it currently stands. Therefore, we invite you to submit a revised version of the manuscript that addresses the points raised during the review process.

We would appreciate receiving your revised manuscript by April 5, 2020. To enhance the reproducibility of your results, we recommend that if applicable you deposit your laboratory protocols in protocols.io, where a protocol can be assigned its own identifier (DOI) such that it can be cited independently in the future. For instructions see: http://journals.plos.org/plosone/s/submission-guidelines#loc-laboratory-protocols

We look forward to receiving your revised manuscript.

Kind regards,

Stephan Doering, M.D.

Academic Editor

PLOS ONE

Reviewers' comments:

Reviewer's Responses to Questions

**Comments to the Author**

1. If the authors have adequately addressed your comments raised in a previous round of review and you feel that this manuscript is now acceptable for publication, you may indicate that here to bypass the “Comments to the Author” section, enter your conflict of interest statement in the “Confidential to Editor” section, and submit your "Accept" recommendation.

Reviewer #1: (No Response)

Reviewer #2: All comments have been addressed

Reviewer #3: (No Response)

2. Is the manuscript technically sound, and do the data support the conclusions?

Reviewer #1: Partly

Reviewer #2: Yes

Reviewer #3: Yes

3. Has the statistical analysis been performed appropriately and rigorously? 

Reviewer #1: No

Reviewer #2: Yes

Reviewer #3: Yes

4. Have the authors made all data underlying the findings in their manuscript fully available?

Reviewer #1: (No Response)

Reviewer #2: Yes

Reviewer #3: Yes

5. Is the manuscript presented in an intelligible fashion and written in standard English?

Reviewer #1: Yes

Reviewer #2: Yes

Reviewer #3: Yes

6. Review Comments to the Author

Reviewer #1: As I have already explained in my original review, authors should perform additional experiments using *real* data to demonstrate their point. Using simulations to generate data is not enough, since in most domains, the the actual distribution differs from the one used in simulations. Further evidence should be provided if this is not the case for psychometric data.

Reviewer #2: Author has addressed all the issues raised by reviewers. Now, paper may be accepted for publication.

Reviewer #3: Given the time series applications and the Monte-Carlo evaluations, it appears that the investigators have met the goal of clarifying the conditions for satisfactory performance of the methods with data typical in psychological and psychosomatic research. The missing and non-equidistant issues certainly could use a fresh look from past procedures.

The Johansen test, in this context, does appear to distinguish between the different types of dynamic systems by estimating the rank of the matrix . In addition, the Johansen procedure provides the maximum likelihood statistical estimation of the parameters from the equations related to the cointegrated systems. Figure 2 is of interest and the lag patterns make sense for the different systems.

Examining the supplemental material, particularly the data sets for S2 to S4, these are really not data sets but simulated examples. The real life types of examples to which these procedures could be applied are descriptively outlined by the authors on lines 309 to 360. It would have been more helpful to have actual data associated with these types of examples to demonstrate the procedures numerically.

7. PLOS authors have the option to publish the peer review history of their article (what does this mean?). If published, this will include your full peer review and any attached files.

Reviewer #1: No

Reviewer #2: Yes: Pushpendra Singh, PhD, Department of ECE, NIT Hamirpur (HP) India

Reviewer #3: No

---

## [Author Response · Author response to Decision Letter 1]

18 Mar 2020

• 197-232: Explanations that the aim of the study can be achieved only with simulated data. 

• The data generated in the study are typical for psychology: non-metric, with missing values... The evaluation method used here is a standard statistical procedure to answer the questions of the study adequately.

• 65-68: Please note that numerous empirical examples with psychometric data are already provided elsewhere.

• The data sets in S2 to S4 provide all simulated data of the study. For instance, in S2 the commands “X=ordinal_10_m(ls$X),Y=ordinal_10_m(ls$Y)” at the end of the code generate 10 cointegrated ordinal measured bivariate systems with length 100 and Beta0=-1, because the set parameters are N=10, T=100, b1=1, b2=1. Changing the parameters to, for example, N=1000 and T=500 generates 1000 cointegrated systems with length 500. With “X1_t=X[,1], Y1_t=Y[,1], CI_1=cbind(X1_t, Y1_t)“ one just gets the first system or with “X1_t=X[,10], Y1_t=Y[,10], CI_1=cbind(X1_t, Y1_t)“ the 10th one. I included the commands print(X) and print(Y) in the codes to demonstrate this (the simulated data can be viewed). Employing a for-loop one can access all N systems successively.

• Please note that the simulated examples in Fig 1 (S1) are metric, S2 to S4 provide simulations codes for generating non-metric (interval, ordinal…) data of the study. 

• The data of the empirical example in Fig 1 are provided as S5_File.csv (S5). The R codes for analyzing the data are included in the appendix (S1: S1_File.R) to show that the Johansen test handles simulated and non-simulated data equally. There are no fundamental differences between these data and other time series from studies outlined on lines 309 to 360.

• Please note that numerous empirical examples with psychometric data are already provided elsewhere (see 65-68) and the aim of the study can be achieved only with simulated data (197-232).

---

## [Decision Letter · Decision Letter 2]

1 Apr 2020

Time series analyses with psychometric data

PONE-D-19-32949R2

Dear Dr. Stadnitski,

We are pleased to inform you that your manuscript has been judged scientifically suitable for publication and will be formally accepted for publication once it complies with all outstanding technical requirements.

With kind regards,

Stephan Doering, M.D.

Academic Editor

PLOS ONE

Reviewers' comments:

Reviewer's Responses to Questions

**Comments to the Author**

1. If the authors have adequately addressed your comments raised in a previous round of review and you feel that this manuscript is now acceptable for publication, you may indicate that here to bypass the “Comments to the Author” section, enter your conflict of interest statement in the “Confidential to Editor” section, and submit your "Accept" recommendation.

Reviewer #2: All comments have been addressed

Reviewer #3: All comments have been addressed

2. Is the manuscript technically sound, and do the data support the conclusions?

Reviewer #2: Yes

Reviewer #3: Yes

3. Has the statistical analysis been performed appropriately and rigorously? 

Reviewer #2: Yes

Reviewer #3: Yes

4. Have the authors made all data underlying the findings in their manuscript fully available?

Reviewer #2: Yes

Reviewer #3: Yes

5. Is the manuscript presented in an intelligible fashion and written in standard English?

Reviewer #2: Yes

Reviewer #3: Yes

6. Review Comments to the Author

Reviewer #2: In this paper, authors present Time series analyses with psychometric data. Author has addressed this reviewer's comments and paper may be accepted for publication.

Reviewer #3: (No Response)

7. PLOS authors have the option to publish the peer review history of their article (what does this mean?). If published, this will include your full peer review and any attached files.

Reviewer #2: Yes: Pushpendra Singh, PhD, Department of ECE, NIT Hamirpur India

Reviewer #3: No

---

## [Editor Report · Acceptance letter]

6 Apr 2020

PONE-D-19-32949R2 

Time series analyses with psychometric data 

Dear Dr. Stadnitski:

I am pleased to inform you that your manuscript has been deemed suitable for publication in PLOS ONE. Congratulations! Your manuscript is now with our production department. 

With kind regards,

on behalf of

Professor Stephan Doering 

Academic Editor

PLOS ONE